

# Deposition of ionic species and black carbon to the Arctic snow pack: Combining snow pit observations with modeling

Hans-Werner Jacobi[1], Friedrich Obleitner[2], Sophie Da Costa[1], Patrick Ginot[1,3], Kostas Eleftheriadis[4], Wenche Aas[5], and Marco Zanatta[1,6]

[1]Institute for Geosciences and Environmental Research (IGE), Univ. Grenoble Alpes / CNRS / Grenoble INP / IRD, Grenoble, France
[2]Institute of Atmospheric and Cryospheric Sciences, University of Innsbruck, Austria
[3]Observatoire des Sciences de l'Univers de Grenoble (OSUG), Univ. Grenoble Alpes / IRD / CNRS / Irstea / Météo France, Grenoble, France
[4]ERL, Institute of Nuclear & Radiological Sciences & Technology, Energy and Safety, NCSR Demokritos, Athens, Greece
[5]Norwegian Institute for Air Research, Kjeller, Norway
[6]Now at: Alfred Wegener Institute (AWI), Helmholtz Centre for Polar and Marine Research, Bremerhaven, Germany

*Correspondence to*: Hans-Werner.Jacobi@univ-grenoble-alpes.fr

**Abstract.** Although aerosols in the Arctic have multiple and complex impacts on the regional climate, their removal due to deposition is still not well quantified. We combined meteorological, aerosol, precipitation, and snow pack observations with simulations to derive information about the deposition of sea salt components and black carbon (BC) from November 2011 to April 2012 to the Arctic snow pack at two locations close to Ny-Ålesund, Svalbard. The dominating role of sea salt and the contribution of dust for the composition of atmospheric aerosols were reflected in the seasonal composition of the snow pack. The strong alignment of the concentrations of the major sea salt components in the aerosols, the precipitation, and the snow pack is linked to the importance of wet deposition for the transfer from the atmosphere to the snow pack. This agreement was less strong for monthly snow budgets and deposition indicating important relocation of the impurities inside the snow pack after deposition. Wet deposition was less important for the transfer of nitrate, non sea salt-sulfate, and BC to the snow during the winter period. The average BC concentration in the snow pack remains small with a limited impact on snow albedo and melting. Nevertheless, the observations also indicate an important redistribution of BC in the snowpack leading to layers with enhanced concentrations. The complex behavior of bromide due to modifications during the sea salt aerosol formation and remobilization in the atmosphere and in the snow were not resolved due to the lack of measurements in aerosols and precipitation.

## 1 Introduction

Aerosols and specifically black carbon (BC) play an important role in the regional climate of the Arctic (Shindell, 2007; Quinn et al., 2007) since they modify the radiation balance of the atmosphere as well as the activation of clouds and reduce the albedo of different cryospheric components like snow and glaciers enhancing the melting of snow and ice after deposition. Arctic aerosols exhibit a pronounced seasonal cycle with high concentrations in winter and early spring and lower values in summer (Law and Stohl, 2007; Quinn et al., 2007; Eleftheriadis et al., 2009). This seasonality is caused by different processes related to emission, transport, and deposition, which undergo seasonal cycles (Law and Stohl, 2007; Croft et al., 2016).



Sea spray, dust, and biogenic aerosol particles are important natural aerosol types in the Arctic. In contrast, Arctic BC stems primarily from regions outside the Arctic (Law and Stohl, 2007). Like in all marine environments, sea salt aerosol (SSA) dominates the atmospheric aerosol burden over the Arctic Ocean and its coastal areas (e.g. Geng et al., 2010; Weinbruch et al., 2012). The production and climatic effects of SSA in the Arctic are expected to change in the future as a result of changes in the sea ice cover and ocean temperatures (Struthers et al., 2011; Zábori et al., 2013). Dust may act as effective ice nuclei in the Arctic (Si et al., 2018) and may have the potential to influence radiative and other properties of mixed-phase cold clouds.

The removal due to deposition controls the lifetime of aerosols and in the Arctic determines the input of the aerosols to the snow and glaciers. In fact, the past atmospheric input has been reconstructed from ice cores in the Arctic (Legrand and Mayewski, 1997; Isaksson et al., 2003; Bauer et al. 2013). Moreover, the deposition of BC to cryospheric components like snow and sea ice also impacts the local and regional climate in the Arctic due to the lowering of the snow albedo and associated albedo feedback processes (e.g. Flanner et al., 2007; Bond et al., 2013; Jacobi et al., 2015). The removal results from wet deposition caused by precipitation and dry deposition of particles, which depend on aerosols size, meteorological conditions, and properties of the atmospheric boundary layer. Despite its importance the deposition of the aerosols to the cryosphere is not well quantified for many polar sites and even the respective contributions of wet and dry deposition are not well known for many compounds (Legrand and Mayewski, 1997; Bauer et al. 2013). As a result, the recommended method to determine dry deposition relies mostly on the calculation of fluxes based on atmospheric composition and an estimated dry deposition velocity (Vet et al., 2014), which shows, however, a large uncertainty. In the case of BC, the calculated deposition varies considerably across models since it depends on the applied assumptions and parameters concerning the size of the aerosols and the mixing state (Bond et al., 2013). For example, the estimated total BC deposition in the Arctic varies between 8 and more than 13 Tg C yr$^{-1}$ with the dry deposition contributing between a few and up to 40 % of the total removal (Liu et al., 2011). Although a comprehensive understanding of microphysics and chemistry related to aging and deposition is essential for a successful simulation of BC concentrations over the Arctic (Liu et al., 2011; Sharma et al., 2013), direct BC deposition measurements are still limited. In addition, the few available measurements of average dry deposition of submicron particles over snow also show a large range from 0.02 to 0.33 cm s$^{-1}$ (Duan et al., 1988; Contini et al., 2010; Grönlund et al., 2002).

While detailed investigations of the chemical properties of aerosols have been performed at Ny-Ålesund based on single particle analysis (e.g. Geng et al., 2010; Weinbruch et al., 2012) similar studies for the composition of the precipitation and the snow pack are currently missing. Moreover, due to the rapid changes in aerosol sources it is urgent to better quantify the fate of different aerosol types in the Arctic. Here, we combine observations in the snow pack and the atmosphere to better constrain deposition processes for major and minor sea salt components and BC around Ny-Ålesund, Svalbard during the winter period. We used meteorological observations to perform detailed physical snow pack modeling. The results of such simulations are to our knowledge for the first time combined with precipitation and atmospheric aerosol measurements to derive chemical profiles and monthly snow budgets related to dry and wet deposition. The calculated profiles are finally compared to physical and chemical snow pack measurements to evaluate the performance of the snow pack model and to improve our understanding of the deposition processes. Variations in the concentrations of multiple




species in aerosols, precipitation, and in the snowpack are used to study the transfer processes from the
atmosphere to the snow for the investigated species.
**2 Methods**
**2.1 Snow sampling**
Sampling of snow was performed in snow pits on two glaciers located approximately 8 and 35 km to the east of
Ny-Ålesund, Svalbard (Fig. 1). The snow was sampled on 30 March 2012 on the Kongsvegen glacier (snow pit
KV, 78.755° N, 13.337° E, ~670 m a.s.l.) and on 13 April 2012 on the Austre Lovénbreen glacier (snow pit AL,
78.871° N, 12.150° E, ~340 m a.s.l.). In both cases, pits were dug down to the ice layer formed during the
summer/fall period in 2011. The sampled snow pack, thus, represented the precipitation accumulated during the
previous winter period. For both snow pits the stratigraphy was established based on visual inspection of the
different layers following international guidelines (Fierz et al., 2009). Snow density was measured using a 250
$cm^3$ triangular snow cutter. Furthermore, duplicate snow samples were collected using 50 mL polypropylene
tubes for either chemical or BC analysis. The tubes with an inner diameter of 2.7 cm were horizontally inserted
into the wall of the snow pit after careful cleaning of the snow surface. The below reported depths of the snow
samples correspond to the center of the tube openings. Furthermore, three fresh snow samples were collected
within 12 h after the end of the precipitation on 24 (two samples) and 29 March at a distance of ~5 km from the
village of Ny-Ålesund, which were analyzed for BC.

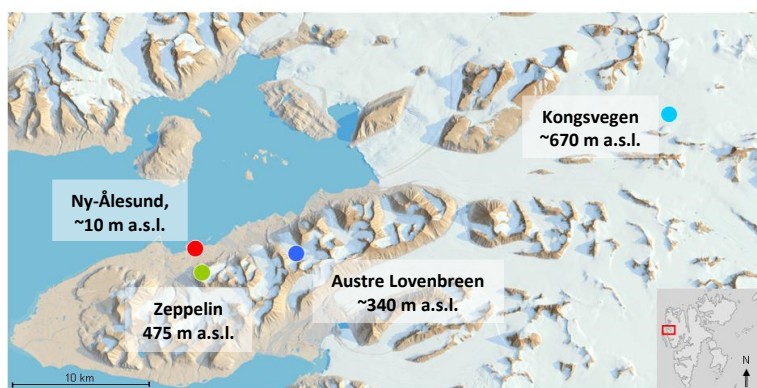

**Fig. 1: 3D map of the Kongsfjord area with Ny-Ålesund (red), Zeppelin Station (green) and the locations of the snow**
**pits on the Kongsvegen (light blue) and the Austre Lovenbreen glaciers (dark blue) indicated (©Norwegian Polar**
**Institute).**
**2.2 Snow analysis**
All samples were stored at -20°C at Ny-Ålesund, transported frozen to Grenoble using isolated boxes and stored
further at -20°C until analysis. Concentrations of the components sodium, potassium, magnesium, calcium,
chloride, sulfate, nitrate, and bromide were determined using ion chromatography in one of the duplicate snow
samples. The samples were filtered using 0.22 µm Acrodisc filters before analysis. Anions and cations were
analyzed with a combined suppressed conductivity Dionex ICS3000 instrument using AS11-HC and CS12
columns. The detection limit is typically below 1 ppb for all measured compounds. The analytical precision is
around 5 % for all ions. Five analyzed samples showed bromide concentrations below the detection limit (3





samples snow pit KV, 2 samples snow pit AL), which were replaced by values of 0.5 ppb for all further
calculations. Non sea salt-sulfate concentrations were calculated according to [nss-sulfate] = [sulfate] – 0.2516 ·
[sodium] (Millero et al., 2008).
Refractory BC (rBC) was determined with a Single Particle Soot Photometer (SP2, Droplet Measurement
Technologies, USA). Details of the analytical procedure are described in Lim et al. (2014). Briefly, the SP2
allows quantifying the mass of single particles using a laser-induced incandescence technique. The instrument
has unity detection efficiency for rBC particles with diameters between 80 and 600 nm, while avoiding
interferences with other inorganic or organic species. The instrument was calibrated using size-selected fullerene
soot (Alfa Aesar Inc., USA). A commercial nebulizer (APEX-Q, Elemental Scientific Inc., Omaha, USA) was
used to transfer the particles from the melted snow to the aerosol phase. The losses during aerosolization were
determined daily using suspensions of Aquadag standards with different mass concentrations resulting in an
average efficiency of 56 %, which was applied to all reported rBC concentrations. Two samples from snow pit
AL showed rBC concentrations below the limit of quantification of 0.03 ppb (Lim et al., 2014), which was used
instead for all further calculations.
**2.3 Meteorological data**
Meteorological parameters have been recorded close to the analyzed snow pit KV by an energy balance station
(KNG8, Karner et al., 2013). The station provided data on air temperature, wind speed, wind direction, short-
and longwave radiation components, and relative humidity. Surface height changes were recorded with an
ultrasonic ranger and allowed deriving accumulation (precipitation). If the measured surface raised more than 1
cm within one hour, the precipitation amount was calculated using a temperature- and wind speed-dependent
parameterization for the density of fresh snow also used in the Crocus model (Vionnet et al., 2012). As a result a
good agreement between observed and simulated increases in snow height was obtained. Cloud cover was
estimated from the ratio between observed and theoretical incoming shortwave radiation using the method
described in Jacobi et al. (2015).
**2.4 Snow pack modeling and snow budgets**
Simulations for the snow pack on the Kongsvegen glacier were performed with the one-dimensional multi-layer
physical snow pack model Crocus (Vionnet et al., 2012; Jacobi et al., 2010a; 2015), which was previously
applied for mass-balance simulations of the glacier (Sauter and Obleitner, 2015). The model solves the surface
mass and energy budgets taking into account physical processes like heat diffusion, transfer of radiation,
densification, sublimation, condensation, and melting. The model is forced using meteorological data like air
temperature, wind speed, relative humidity, precipitation rate and phase, incoming direct and diffuse short-wave
radiation, incoming long-wave radiation, and cloud cover. The forcing data for the period September 2011 to
March 2012 were generated from observations at the energy balance station KNG8. The model was initiated
with an ice layer set to a temperature of 0°C. The output refers to multiple homogeneous horizontal layers that
are established according to snowfall events and undergo transformation related to a metamorphism scheme. The
model delivers physical properties of each snow layer including thickness, density, temperature, structure
parameters and date of accumulation. According to the simulations the oldest conserved snow pack layer was
deposited on 30 October 2011. The simulations deliver accumulation dates for each simulated layer, which were





used to divide total snow budgets into monthly budgets for October 2011 and March 2012. However, the
incomplete October budgets were not used for further analysis. Concentrations and densities of the simulated
snow layers were used to calculate monthly budgets for each impurity.
**2.5 Atmospheric concentrations**
Atmospheric concentrations $C_{atm}$ of ionic species in the aerosols were measured at Zeppelin Station at an altitude
of 475 m a.s.l.. The aerosols were collected using a three stage filterpack sampler with no size cut off (Aas et al.,
2013). The data were downloaded from the EBAS database (ebas.nilu.no) for the period from 30 October 2011
to 29 March 2012 as daily averages. Concentrations of atmospheric BC corresponding to equivalent BC (eBC)
were determined at the Zeppelin station using the 880 nm channel BC values recorded by a 7-wavelengths AE31
aethalometer (Eleftheriadis et al., 2009) using updated parameters for the absorption and attenuation according to
Backman et al. (2017) and Zanatta et al. (2018). Daily averages were used for further calculations (see
Supplementary Material, Fig. S1). The eBC time series includes 23 missing values (7 days in November, 16 days
in December) and 19 values below the detection limit (BDL) of 7 ng m$^{-3}$ in the analyzed period. While the
missing values were replaced by the monthly averages, the impact of using either the maximum or minimum
value (7 or 0 ng m$^{-3}$) to replace the values BDL remained small for calculated monthly averages (< 0.8 ng m$^{-3}$
and < 7%). Therefore, all values BDL were replaced by 3.5 ng m$^{-3}$ equal to half of the detection limit before
further calculations.
**2.6 Dry and wet deposition**
Dry deposition of particles ($D_{dry}$; g m$^{-2}$) was calculated using Eq. (1),
$D_{dry} = C_{atm} \cdot v_d \cdot t \cdot 10^{-6}$                                                                                    (1)
with the deposition velocity $v_d$ in m s$^{-1}$, the atmospheric concentration of the aerosols $C_{atm}$ in µg m$^{-3}$, and the
averaging time $t$ for the atmospheric measurements (here 24 h). While wind speed- and particle size-dependent
parameterizations for $v_d$ are available (e.g. Zhang et al., 2001), the use of single values of $v_d$ to estimate dry
deposition fluxes from aerosol concentrations is still standard. Here, we used a high $v_d$ value of 1 cm s$^{-1}$ for the
ionic species assuming that they were mainly associated with coarse sea salt aerosols (Zhang et al., 2001). Thus,
we consider the estimated dry deposition of these components as an upper limit. The gas-phase species nitric
acid (HNO$_3$) and sulfur dioxide (SO$_2$) also contribute to the total dry deposition of nitrogen and sulfur, but are
not included in the estimates here. Since bromide was not determined in the aerosols, the standard sea water ratio
and observed sodium concentrations were used to estimate the dry deposition of bromide. A smaller $v_d$ of 0.1 cm
s$^{-1}$ was used for BC, which corresponds to a typical global annual mean in many models (Wang et al., 2011) and
is within the range of observed deposition velocities of sub-micron particles over snow (Duan et al., 1988;
Contini et al., 2010; Grönlund et al., 2002). The total and monthly dry deposition was calculated as the sum of
the daily deposition either for the full period or for each month from November 2011 to March 2012.
Wet deposition ($D_{wet}$; g m$^{-2}$) was calculated using observed precipitation amounts ($P$; L m$^{-2}$) and chemical
concentrations ($C_{precip}$; mg L$^{-1}$) of the precipitation collected at Ny-Ålesund close to sea level using Eq. (2):
$D_{wet} = C_{precip} \cdot P \cdot 0.001$                                                                                    (2)
Major sea salt components and nitrate were determined using ion chromatography in precipitation samples
collected on a weekly basis using a bucket funnel system in summer and a snow sampler in winter (Kühnel et al.,





2011; Aas et al., 2013). The data downloaded from the EBAS database (ebas.nilu.no) were used without further
correction, although the bulk sampler likely collected also gaseous compounds and particulate material due to
dry deposition. Especially in periods with high wind speed, the bulk collector may also catch large sea spray
aerosols. However, the exact contribution of dry deposition to the here calculated wet deposition is difficult to
quantify since it depends on the frequency of rain events and episodes with elevated sea salt aerosols. The total
and monthly wet deposition was calculated as the sum for the period from 31 October 2011 to 1 April 2012 and
for each month (except October). No measurements of bromide and BC in the precipitation are available. For
bromide, wet deposition was estimated from the wet deposition of sodium applying the standard sea water
composition according to $D_{wet}$(bromide) = 0.00624 · $D_{wet}$(sodium) (Millero et al., 2008). Wet deposition of BC
was estimated according to the scavenging scheme proposed by Sharma et al. (2013). The change in atmospheric
BC concentration $\Delta[BC]$ was estimated using the BC concentration $[BC]$, the scavenging coefficient ($R$; $m^2$ $kg^{-1}$
), the precipitation rate ($P_t$, L $m^{-2}$ $s^{-1}$) and the time step ($\Delta t$, s) according to Eq. (3):
$\Delta[BC] = [BC] \cdot R \cdot P_t \cdot \Delta t$ (3)
We used a scavenging coefficient of $R = 5 \cdot 10^{-3}$ $m^2$ $kg^{-1}$ as recommended by Sharma et al. (2013). Since all
scavenged atmospheric BC will be mixed into the accumulated weekly snowfall (($P_t \cdot \Delta t$), the BC concentration
in the snow $[BC]_{snow}$ in ppb is calculated according to Eq. (4), where $h$ denotes the depth of the atmospheric
column affected by scavenging:
$[BC]_{snow} = 10^{-9} \cdot [BC] \cdot h \cdot R$ (4)
Since typical top heights of clouds in the wintertime Arctic are on the order of 4500 m (Intrieri et al., 2002) we
used $h$ = 4500 m for the tropospheric column concerned by scavenging. BC concentrations in snow were
calculated only for weeks with precipitation recorded at Ny-Ålesund and using the observed weekly precipitation
rates. For further calculations, we used monthly average BC concentrations in the snow due to scavenging to
derive monthly wet deposition according to Eq. (2). Finally, total wet deposition corresponds here to the sum of
the monthly wet deposition from November to March.
**3 Results and discussion**
**3.1 Physical properties of the snow pack**
The stratigraphy and densities for the two analyzed snow pits are shown in Fig. 2. The investigated snow layers
comprised depths down to -263 cm below the surface for snow pit KV and -195 cm for snow pit AL reaching in
both cases the surface of the ice layer formed during the previous summer. While both snow pits showed a
typical increase in density from the surface to the deeper layers, the variability in terms of grain types and layer
structures was higher for snow pit AL. At an altitude of 670 m a.s.l. the high wind speeds at snow pit KV led to
the formation of several wind-packed layers. The impact of significant melting was not identified in the snow pit
KV, although the recorded temperatures reached several times values above or close to the melting point (see
Supplementary Material, Fig. S2). In contrast, at an altitude of 340 m a.s.l. melting events were more apparent in
snow pit AL, which exhibited several melt freeze crusts probably due to warmer periods in November 2011 and
January 2012 accompanied by air temperatures above 0°C and large amounts of rain at sea level in Ny-Ålesund.
The stronger impact of melting in the snow pit AL was confirmed by the chemical composition. The ratio of
magnesium to sodium has been proposed as a melt indicator (Iizuka et al., 2002; Virkkunen et al., 2007; Ginot et





al., 2010) with lower ratios caused by the preferential removal of magnesium due to percolating water. While the
average magnesium to sodium ratios were around 0.12 in both snow pits, in snow pit AL the variability was
higher and minimum values lower. Smallest ratios were encountered in layers deposited in November, January,
and March corresponding to the months with elevated air temperatures. Nevertheless, in both snow pits the ratios
did not reach the small ratios as observed in ice cores from Svalbard (Iizuka et al., 2002; Virkkunen et al., 2007).
Therefore, a certain redistribution of the impurities probably occurred in the snow pack due to melting, but not a
complete removal. While the impact was stronger on the Austre Lovenbreen glacier, the overall budgets of both
snow pits seemed not to be influenced.

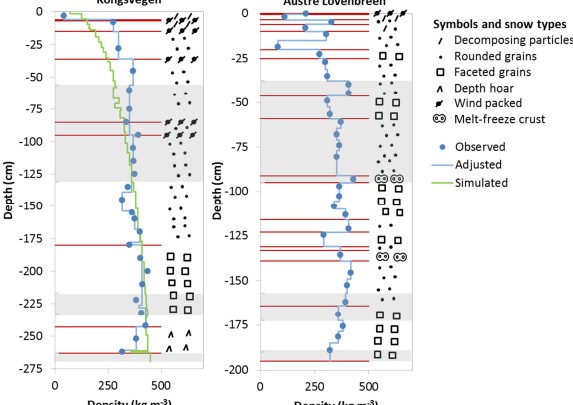

**Fig. 2: Snow stratigraphy observed in the snow pits KV (left) and AL (right). Note the different depth scales. Snow**
**types are indicated on the right side using the classification recommended by Fierz et al. (2009) with layers separated**
**by horizontal red lines. Blue circles indicate observed snow densities, vertical blue lines correspond to snow densities**
**adjusted to observed and sampled layers. The green line shows snow densities simulated with the Crocus model for**
**snow pit KV. The alternating shaded and non-shaded areas correspond from the top to the months March to October.**
Together with the stratigraphy, full snow density profiles were established for both pits as shown in Fig. 2.
According to these profiles, the total accumulation amounts to 943 and 667 mm snow water equivalent (SWE)
for the snow pits KV and AL. The accumulation in the pit KV was close to the maximum observed in the years
2007 to 2009 at altitudes above 600 m on the Kongsvegen glacier (Forsström et al., 2013). The observed
accumulation of precipitation at Ny-Ålesund close to sea level corresponds to a value of 278 mm for the period
from 31 October 2011 to 01 April 2012. The gradient in accumulation between the snow pits KV, AL, and Ny-
Ålesund was slightly higher than 30 % per 100 m altitude increase and is, thus, close to accumulation gradients
previously applied for the nearby Midre Lovénbreen and Austre Brøggerbreen glaciers (Hodson et al., 2005).
Crocus model results obtained for the snow pit KV were used here for further analysis. The snow pack simulated
for 29 March 2012 consists of 50 layers with varying densities covering a total depth of -269.4 cm (Fig. 2). The
densities below -80 cm are well represented by the model, but the densities between -10 and -80 cm are
underestimated compared to the observations. While density measurements using a cutter suffer from a
systematic overestimation (Proksch et al., 2016), a strong bias in simulated snow densities in the top layers is
common for snow models applied to polar regions (e.g. Groot Zwaaftink et al., 2013). Despite the differences in
the top layers, the simulated total SWE of 937 mm is in excellent agreement with the observed SWE. Shaded
areas in Fig. 2 indicate the different layers deposited in the months from October 2011 to March 2012 according



to the simulations. Only complete identified snow layers were attributed to specific months. Assuming a constant
linear decrease in accumulation from the snow pit KV to AL, monthly layers were also attributed to snow pit AL
with linearly interpolated depth ranges using the ratio of the total snow heights of both pits (Fig. 2).

**3.2 Impurity profiles in the snow pack**

Co-located impurity profiles were established for each measured compound combining the measured
concentrations with either the observed stratigraphy for both snow pits or the simulated stratigraphy for snow pit
KV. Figure 3 shows as examples the observed sodium and rBC concentrations as well as the profiles. Some
common features can be identified for sodium in the upper part of both snow pits. Snow pit KV showed three
layers with elevated concentrations: a first peak in the March layer around -25 cm, and a double peak in the
February layer with a maximum around -60 cm and a broad maximum below -100 cm (Fig. 3a). A comparable
pattern was found in snow pit AL with the strongest peak in the March layer at -20 cm and two maxima in
February at -40 cm and a broad peak between -60 and -90 cm (Fig. 3b). However, the relative strengths of the
peaks are different in the two pits. While these differences may be caused by site-dependent deposition fluxes,
they may also be related to melt-water formation and percolation, which likely had a stronger impact on snow pit
AL. This may have contributed also to the higher variability of sodium in the lower part of the snow pit AL
compared to KV. Moreover, peaks may have been missed at KV because of the larger spacing between the
samples.
The lowest concentrations of all studied impurities were found for rBC. Average rBC concentrations differed by
a factor of two between the two location with 0.6 ppb at KV (Fig. 3c) and 1.2 ppb at AL (Fig. 3d). The average
concentration at AL is in good agreement with the average rBC concentration of (1.4 ± 0.2) ppb for the snow
pack accumulated during the winter 2012/2013 on the Brøggerbreen glacier at 300 m altitude (Sinha et al.,
2018). However, the concentrations are lower than elemental carbon (EC) measured with a thermo-optical
method in samples from the Kongsvegen glacier above 600 m a.s.l, where Forsström et al. (2013) found median
concentrations of 1.4, 4.2, and 3.8 ppb in April 2007, 2008, and 2009. It should be noted here and during all
further discussion, that the comparison of the different quantities EC, eBC, and rBC introduces additional
uncertainties. For example, a previous comparison of EC and rBC in snow samples from Greenland indicated a
mean ratio of two (Lim et al., 2014), while atmospheric rBC and eBC measurements in Ny-Ålesund are more
consistent if adjusted absorption properties are used (Zanatta et al., 2018). These uncertainties may partly explain
the differences in the determined concentrations. While an overall decrease of atmospheric BC concentrations in
the Arctic (Sinha et al., 2017) potentially contributed to lower concentrations in 2012 compared to the years
2007 to 2009, the impact is difficult to analyze due to large inter-annual meteorological variabilities impacting
the transport of BC to the Arctic (Eleftheriadis et al., 2009; Hirdmann et al., 2010) occasionally leading to high
pollution events in wintertime (e.g. Winiger et al., 2012). Melting processes probably caused the higher
variability of BC in the AL pit since insoluble particles show enrichments at the snow surface generating layers
with enhanced concentrations in the Arctic (Forsström et al., 2013).



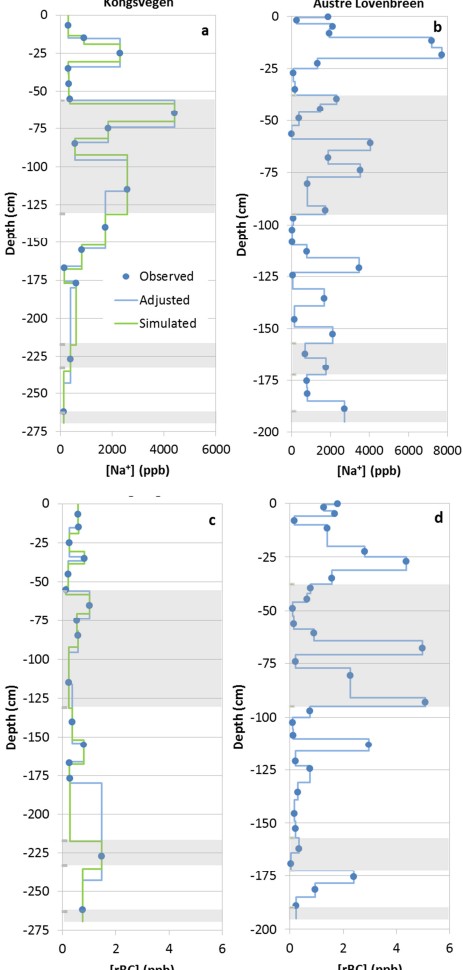


**Fig. 3: Sodium (top) and rBC concentrations (bottom) in the snow pits KV (left) and AL (right). Blue circles indicate**
**observed concentrations and blue lines correspond to concentrations adjusted to the observed stratigraphy. The green**
**line shows concentrations adjusted to the simulated stratigraphy for the snow pit KV. The alternating shaded and**
**non-shaded areas correspond from the top to the months March to October.**
**3.3 Wintertime snow budgets and deposition of ionic compounds**
Total snow budgets of all measured compounds for the two snow pits were calculated using three different
approaches: (i) simple budgets were determined by multiplying the average concentrations by the total SWE; (ii)
adjusted budgets were calculated from the interpolated density profile shown in Fig. 2 and co-located
concentration profiles like in Fig. 3; (iii) for snow pit KV simulated budgets were obtained by combining the
simulated density profiles with simulated concentration profiles. All calculated budgets are summarized in Fig.
4, which also shows the observed wet deposition at Ny-Ålesund and the estimated total dry aerosol deposition
for the period from 31 October 2011 to 29 March 2012. According to the meteorological records of the
Norwegian Meteorological Service (eklima.met.no) no further precipitation occurred between 29 March and 15
April 2012 and the total wet deposition can, thus, be compared to the budget of the snow pit AL.





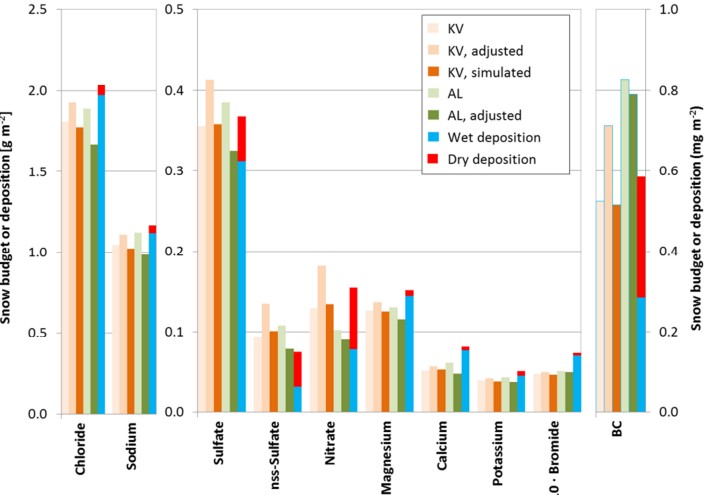

**Fig. 4: Snow budgets of sea salt components, nss-sulfate, nitrate and bromide (left and middle) and BC (right) for the snow pits KV (Kongsvegen, brown) and AL (Austre Lovénbreen, green) for October 2011 to March 2012 according to different calculation methods: simple budgets from average concentrations and total SWE (light colors), adjusted budgets using co-located concentration and density profiles (middle colors), and simulated budgets for KV using interpolated concentration and simulated density profiles (dark colors). Also shown is the total deposition as the sum of observed wet deposition at Ny-Ålesund (blue) and estimated dry particle deposition (red). For bromide the wet deposition was estimated from the standard sea water ratio and all numbers are multiplied by ten. For BC the snow budgets correspond to rBC, while the wet deposition was estimated for the KV snow pit. The BC deposition corresponds to eBC. See text for further details.**

Due to errors of the manual snow density measurements, the chemical analysis, and the extrapolation of the density and concentration profiles, it can be assumed that differences in the budgets below 20 % as obtained for chloride, sodium, magnesium, calcium, and potassium are not significant. The spatial variability of snow concentrations at a scale of meters can be even larger (e.g. Svensson et al., 2013). Thus, the total snow budgets for the pits KV and AL reveal a consistent picture for the sea salt components chloride, sodium, magnesium, potassium, and bromide (Fig. 4). For these species neither the method for the calculation of the total budgets, the location, the altitude, nor the accumulation led to significant differences in the observed total budgets. This is consistent with recent observations revealing characteristic patterns of aerosol concentrations along Svalbard glaciers including the Kongsvegen demonstrating consistent formation, transport, and exchange processes between the atmosphere and the snow (Spolaor et al., 2017).

If post-depositional processes are negligible, the total snow budgets of the impurities correspond to the input due to the sum of the wet and dry deposition. Based on the comparison of the total snow budgets with the observed wet deposition, the estimated dry deposition are evaluated for the different impurities. The total snow budgets of chloride, sodium, magnesium, and potassium agree well with the observed wet deposition at Ny-Ålesund with differences smaller than 20 % for the period from October 2011 to March/April 2012. However, the recorded wet deposition also includes variable contributions from dry deposition since the precipitation samples were collected with an open bucket instrument (Kühnel et al., 2011). Nevertheless, the estimated dry deposition corresponds to less than 5 % of the wet deposition of chloride, sodium, and magnesium and reaches a maximum of 14 % for potassium (Fig. 4). Subtracting the nss-sulfate from the total sulfate shows that the dry deposition of sulfate with marine origin also corresponds to less than 5 % of the total wet deposition. Since the estimated dry deposition is considered as an upper limit, it can be assumed that its contribution for the total snow budget on the



Kongsvegen and Austre Lovenbreen glaciers during the period November 2011 to April 2012 remained small for
chloride, sodium, magnesium, potassium, and sea salt sulfate. The estimated wet deposition for bromide based
on sodium concentrations and the standard sea water ratio leads to an overestimation of more than 40 %
compared to the observed bromide in the snow pack (Fig. 4). This demonstrates that sea salt bromide is
undergoing important modifications during the formation of sea salt aerosols, in the atmosphere, or after
deposition (see Sect. 3.7).
Like for the sea salt components, a good agreement between the KV snow budget of nitrate and nss-sulfate and
the total deposition during the period from October 2011 to April 2012 is found. For these two compounds the
observed wet deposition at Ny-Ålesund remains significantly below the snow budget, while the missing fractions
are largely compensated by the estimated dry deposition. For nitrate, the dry deposition is comparable to the wet
deposition, whereas for nss-sulfate dry deposition even dominates de snow budget. The adjusted budgets of the
snow pit AL show ~50 % less nitrate and ~40 % less nss-sulfate compared to KV (Fig. 4), which may be related
to the spatial variability of the dry deposition of the two species.

**3.4 Wintertime snow budgets and deposition of BC**

Regarding the snow budgets, the different average rBC concentrations are partly compensated by the different
accumulation for the two snow pits. The simple, adjusted, and simulated snow budgets vary between 0.51 and
0.71 mg m$^{-2}$ for KV and 0.79 and 0.83 mg m$^{-2}$ for AL and are, thus, between 10 and 60 % higher at AL
compared to KV. Albeit the difference between EC and rBC (see Sect. 3.2), it appears that the derived rBC
budget for KV remain well below the EC budgets for 2007 to 2009 for the Kongsvegen glacier (Forsström et al.,
2013). In contrast, the AL budget is somewhat above the rBC budget for the Brøggerbreen glacier of 0.49 mg m$^{-2}$
determined in April 2013 (Sinha et al., 2018). These differences probably also correspond to the inter-annual
variability of BC as reflected in the atmospheric observations (Eleftheriadis et al., 2009).
The BC deposition derived here only relies on estimated values for scavenging and dry deposition velocities.
Nevertheless, the monthly averages of the estimated BC concentrations in fresh snow due to scavenging varying
from 0.2 ppb in October 0.8 ppb in March. The March average is in very good agreement with the three fresh
snow samples collected in March 2012, which showed rBC concentrations of 0.47, 0.72, and 0.97 ppb,
respectively. Moreover, these estimated concentrations are in agreement with average eBC concentrations in
fresh snow samples collected in April and November in the years 2012 to 2017 on the glaciers surrounding Ny-
Ålesund ranging from 0.9 to 2 ppb (Gogoi et al., 2018). Finally, Noone and Clarke (1988) proposed a
dimensionless scavenging ratio (SR) corresponding to the quotient of the atmospheric and snow concentration of
BC. Using the observed atmospheric BC and the estimated BC concentrations in snow we derive SRs of
approximately 30, which agree with the lower end of SRs for the Ny-Ålesund area (Gogoi et al., 2016; 2018).
The anti-correlation between accumulation and the average BC concentrations in the two snow pits points to an
important contribution of dry deposition, which is in agreement with the estimated dry and wet deposition of BC.
The dry deposition of eBC (Fig. 4) derived with a deposition velocity of 0.1 cm s$^{-1}$ corresponds to approximately
half of the observed rBC budget at KV and is, thus, somewhat higher than the contribution due to wet deposition.
Despite the overall uncertainty related to the simplified methods for the estimation of the BC deposition, the
difference between the snow budgets and the total deposition remain below 25 % for the KV and below 45 % for
the AL snow pit. This important contribution of dry deposition is in contrast to wet and total deposition reported





for Ny-Ålesund for the winter 2012/2013 based on rBC measurements in falling snow and in the snow pack
(Sinha et al., 2018). From these observations it was concluded that the dry deposition of rBC remained
negligible. However, the authors also reported rBC fluxes at 300 m altitude on the Broggerbreen glacier, which
were twice as high as in Ny-Ålesund. While Sinha et al. (2018) claim that this increase is mainly due to the
higher accumulation on the glacier, additional dry deposition at higher altitudes cannot be excluded. Moreover,
the potential contamination of the snow pack close to Ny-Ålesund due to local power generation or a potential
mismatch between the budgets of the falling snow and the snow pack due to the removal by blowing snow were
not considered.
Previous model studies have indicated that BC in the Arctic is primarily removed through wet deposition (e.g.
Flanner et al., 2007; Wang et al., 2011). However, in the models the dry deposition velocity of BC was often
reduced to improve the simulated atmospheric concentrations of BC. Moreover, the here observed wintertime
deposition may not be extrapolated to the entire Arctic since the BC deposition depends on multiple factors like
air mass transport, aging processes of atmospheric BC particles, and ice nucleation (e.g. Sharma et al., 2013; Liu
et al., 2011; Vergara-Temprado et al., 2018)
**3.5 Comparison of monthly snow budgets and deposition**
To derive a higher temporal resolution of the snow budgets monthly snow budgets were calculated from layers
deposited in each month between November 2011 and March 2012. The monthly budgets are further compared
to monthly wet and dry deposition. Each weekly wet deposition was attributed to the month with the largest
overlap in time to derive the monthly wet deposition, while the monthly dry deposition was calculated from the
daily dry deposition. Monthly total deposition was calculated as the sum of the corresponding wet and dry
deposition. Figure 5 shows as example the results for sodium, nitrate, and BC. Results similar to sodium were in
general obtained for the other sea salt components. The dominating role of wet deposition for sodium and other
sea salt components and the larger contribution of dry deposition for nitrate and BC are also reflected in the
monthly budgets. For the months with recorded precipitation at Ny-Ålesund, the wet deposition of sea salt
components largely dominates the total deposition. This is in contrast to nitrate and BC, which show several
monthly budgets with higher values for dry than wet deposition.
The generally good agreement between the total budgets of the two snow pits and the wet and dry deposition
(Fig. 4) is only partly confirmed by the monthly budgets shown in Fig. 5. For example, the monthly budgets of
sodium show a much more pronounced variability at KV compared to AL. In contrast, the monthly total
deposition shows a very low value for December due to the lack of wet deposition observed at Ny-Ålesund and
no clear cycle for the remaining months. Similar results are obtained for other sea-salt components. In general,
differences are caused by multiple reasons related to uncertainties in the forcing data, in the model results as well
as in the spatial variability of the observations in the snow, wet deposition, and aerosol concentrations.
Moreover, post-depositional processes modifying the derived monthly snow budgets like blowing snow or
melting processes are currently not taken into account in the simulations. Nevertheless, it appears that for
compounds with a larger contribution of dry deposition the agreement between snow budgets and total
deposition is somewhat better like in the cases of nitrate and BC (Fig. 5).





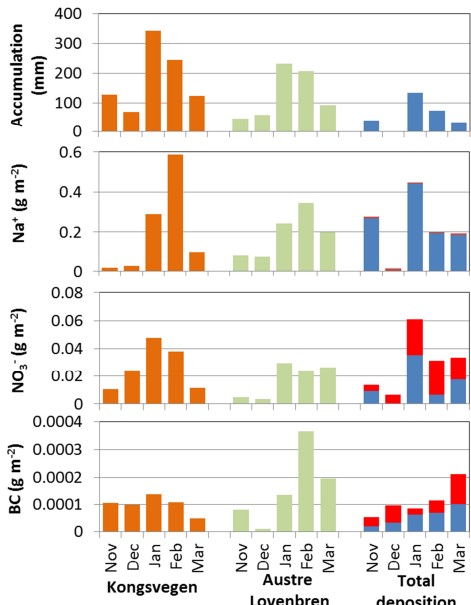

**Fig. 5: Monthly accumulation and snow budgets for sodium, nitrate, and BC for the snow pits KV (Kongsvegen,**
**orange) and AL (Austre Lovénbreen, green) according to the simulated profiles. Also shown is the observed**
**accumulation at Ny-Alesund (top, blue) and the total deposition divided into wet (blue) and dry (red) deposition. For**
**sodium and nitrate the wet deposition was measured at Ny-Ålesund and the wet deposition for BC was estimated from**
**scavenging. The BC snow budgets correspond to rBC, while the wet and dry depositions correspond to eBC.**
**3.6 Variations of the chemical composition of snow, aerosols, and precipitation**
Variation diagrams showing the concentrations of two trace compounds are often exploited to determine
common sources or processes acting upon the correlated species. Here, the ratios of concentrations in the
atmosphere, the precipitation, and the snow are used to study the transfer processes from the atmosphere to the
snow. Figure 6 shows the variation of chloride vs sodium for the period from October 2011 to April 2012 in
aerosols at Zeppelin Station, in the precipitation at Ny-Ålesund, and in the snow pits KV and AL. Most of the
chloride-to-sodium ratios in the aerosols are close to the standard sea water ratio (Millero et al., 2008) indicating
that in the marine environment around Ny-Ålesund the composition of the aerosols is dominated by sea salt.
Some aerosol samples show dechlorination likely caused by the replacement of chlorine ions due to the uptake of
sulfuric and nitric acid (Keene et al., 1998). Figure 6 demonstrates further that the impact of the dechlorination
becomes visible only during periods with low atmospheric loading of sea salt aerosols with less than 1 μg m$^{-3}$
sodium. Such a dechlorination, however, is masked in the precipitation and snow samples, which were all close
to the standard sea water ratio. In summary, in wintertime the composition of the majority of the aerosols as well
as the precipitation around Ny-Ålesund is dominated by sea salt aerosols confirming previous studies of
individual aerosol particles (Weinbruch et al., 2012). This leads to the same chemical imprint in the snow pack.
Magnesium and potassium show a similar variation with sodium (not shown), except that some aerosol samples
showed elevated potassium concentrations possibly due to potassium-enriched dust particles from soils.
Nevertheless, the variation in the precipitation samples confirms that the contribution of dry aerosol deposition
remained small also for these compounds with respect to the overall observed budget of the snow pits.



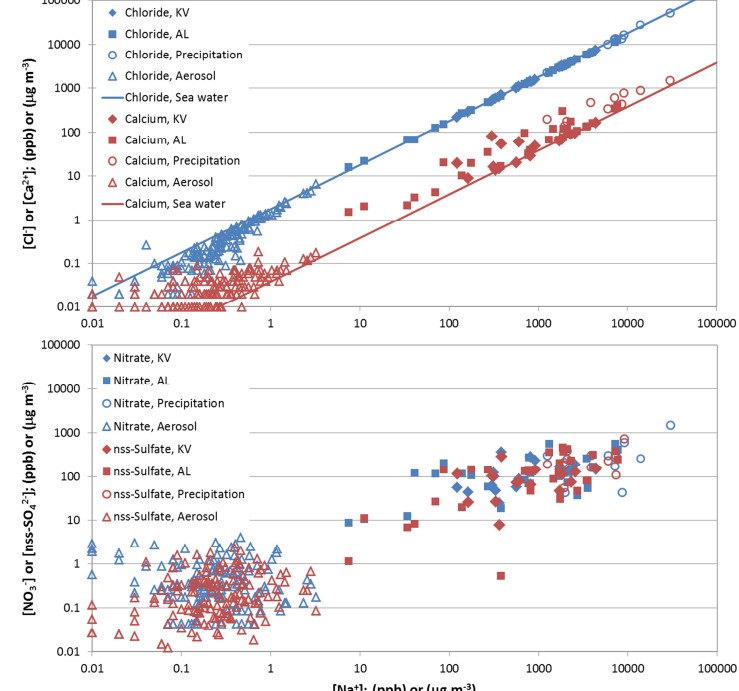


**Fig. 6, upper panel: Chloride (blue) and calcium (red) concentrations vs. sodium concentrations in snow pit KV (filled**
**diamonds) and AL (filled squares), in precipitation (open circles) and in aerosols (open triangles). Aerosol**
**concentrations are in µg m$^{-3}$ and are derived from measurements at Zeppelin Station. The lines indicate the standard**
**sea water ratio. Lower panel: Same as top, but for nitrate (blue) and nss-sulfate (red).**
Calcium shows a different behavior compared to the other major sea salt components with a significant
enrichment of calcium in the aerosols as well as in the precipitation, which also causes calcium-to-sodium ratios
above standard sea water in a large number of snow samples (Fig. 6). Such an enrichment in the Arctic may be
attributed to calcium-rich aerosols originating from soils (Toom-Sauntry and Barrie, 2002; Geng et al., 2010;
Jacobi et al., 2012; Weinbruch et al., 2012) although local aerosol formation was probably limited due to the
extended snow cover. The in- or below-cloud scavenging of dust particles likely contributed to the transfer of the
elevated calcium concentrations from the aerosols to the precipitation and to the snow.
Like demonstrated in previous studies (e.g. Jacobi et al., 2012), nitrate and nss-sulfate in aerosols, in the
precipitation, and in the snow do not exhibit a constant ratio compared to sodium (Fig. 6). On average, highest
and lowest ratios are found in the aerosols and in the precipitation with the average snow pack ratio in between
these values. This confirms that the nitrate and nss-sulfate in the snowpack can be attributed to a mixture of wet
deposition and dry deposition of aerosols. Although in wintertime the reactive nitrogen budget is dominated by
particulate nitrate (Hara et al., 1999), a further dry deposition of gas phase species to the snow is possible, which
may be even more important than the aerosol deposition (Björkman et al., 2013).
Due to the different sources of BC and sodium (long-range transport vs local or regional formation of sea salt
aerosols), no consistent BC-to-sodium variation is found in the aerosols (Supplementary material, Fig. S3).
Similarly, the variation of BC in the snow pits is also independent of the sodium concentrations. Since BC
particles are preferentially coated by organic matter or sulfate (Liu et al., 2011), atmospheric BC shows a linear





relationship to nss-sulfate resulting in a correlation coefficient $R^2$ of 0.60 (Fig. 7). In the snowpack, the rBC-to-
nss-sulfate ratios are less consistent and the average ratio is almost one order of magnitude smaller than in the
atmosphere. Despite the different measurement techniques for BC in the aerosols and in the snow, the lower BC-
to-nss-sulfate ratio in the snow can only partly be explained by the different measurement methods. Different
ratios in the snow may be caused by the smaller contribution of wet to total deposition of BC as compared to nss-
sulfate (Fig. 4). Moreover, the AL snow pit shows a higher variability in the BC-to-nss-sulfate variation than the
KV snow pit (Fig. 7) indicating that redistribution of the impurities caused by melting probably also impacted
BC and nss-sulfate.

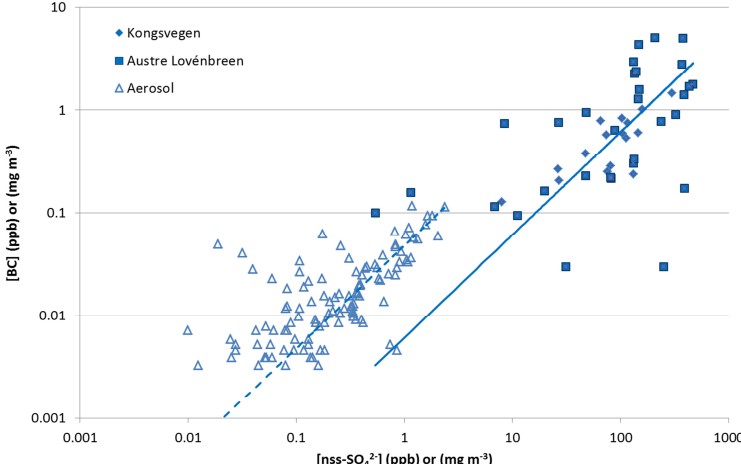

**Fig. 7: BC vs. nss-sulfate concentrations in snow pit KV (filled diamonds) and AL (filled squares), and in aerosols**
**(open triangles). The lines calculated by linear regression are forced through the origin and separated for the aerosol**
**and the snow samples. BC in the snow pits corresponds to rBC, while the BC in aerosols corresponds to eBC.**
**3.7 Bromide in the snowpack**
Since no bromide concentrations in the aerosols and in the precipitation are available, the ratio between bromide
and sodium is shown in the form of profiles for the snow pits KV and AL (Fig. 8). The ratio between the overall
bromide and sodium budgets varies between 0.0045 for KV and 0.005 for AL and is, thus, below the standard
sea water ratio of 0.00624 (Millero et al., 2008). Only distinct layers show enrichments of bromide (Fig. 8).
Multiple photochemical processes occur in the sea ice-snow-atmosphere system of the Arctic acting upon the
variation between bromide and sodium (Simpson et al., 2007; Jacobi et al., 2012). On solid surfaces (aerosols,
snow, sea ice) bromide can be transformed into volatile bromine compounds that are released to the atmosphere
and are subsequently deposited. Therefore, bromide can be depleted already in the sea salt aerosols generated
over sea ice, which would cause a wet and dry deposition flux lower than estimated based on the standard sea
water ration, or it can be diminished in the surface snow after deposition (Jacobi et al., 2012) explaining the
average bromide-to-sodium ratios below the sea water ratio in both snow pits. Nevertheless, since the released
bromide is subsequently deposited, a snow pack with layers enriched in bromide is also possible depending on
the dominating influence of the release vs the additional deposition of bromide (Simpson et al., 2007). This can
also explain the contrasting results found on top of the Holtedahlfonna glacier, located approximately 40 km to
the Northeast of Ny-Ålesund in April 2012. Spolaor et al. (2013) reported that the snow pack was highly
enriched in bromide with only a few samples close to the sea water ratio potentially caused by the additional





deposition of bromide after release from sea ice-covered areas of the Arctic Ocean. It is well known that such
activation of bromide mainly occurs in springtime after polar sunrise explaining the low bromide budgets in the
KV and AL snow pits in the winter period. Since the bromine activation over sea ice also leads to a significant
destruction of tropospheric ozone (Jacobi et al., 2010b), the ozone record at Zeppelin Station may be used as a
proxy for the impact of bromine-rich air masses at Ny-Ålesund and the surrounding area. The ozone
concentration during the entire period covered by the snow pits remained above 35 ppbV and dropped to ~20
ppbV on the afternoon of 29 March (data accessible at ebas.nilu.no) indicating that the influence of bromine-
enriched air remained small in spring 2012 before the sampling of the snow pits. Finally, meteorological
conditions in Ny-Ålesund are less sensitive to the sea ice extent than other locations in Svalbard according to a
study of precipitation trends (Osuch and Wawrzyniak, 2016). Thus, the distinct peaks in the bromide-to-sodium
in the snow pit AL, which occurred in layers attributed to early March, late-February, and late January are
probably not caused by additional deposition as a consequence of bromine explosion events, but are possibly
related to the influence of melting. Accordingly, these high ratios are not produced by enhanced bromide, but by
low sodium concentrations. In summary, the potential use of bromide enrichment as proxy for the bromine
activation and, thus, sea ice extent as developed by Spolaor et al. (2013) for the Holtedahlfonna record appears
rather limited for the snow pack and glaciers around Ny-Ålesund due to a restricted impact of sea ice conditions
in this region.

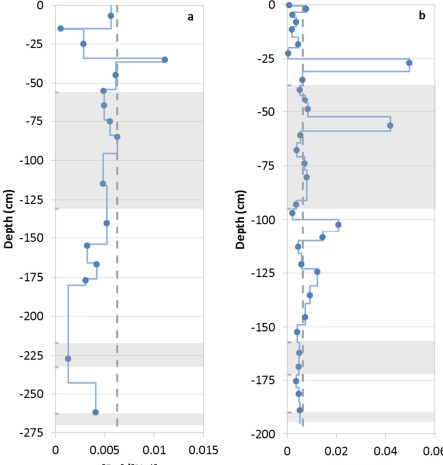

Fig. 8: Bromide-to-sodium ratio in the snow pits KV (left) and AL (right). Blue circles indicate observed ratios; blue
lines correspond to ratios adjusted to the observed stratigraphy. The alternating non-shaded areas correspond to the
months November, December, January, February, and March (from the bottom). The vertical dashed grey lines
indicate the standard sea water ratio.
**4 Conclusion**
The chemical composition of aerosols, precipitation, and the snow pack was analyzed for Ny-Alesund, Svalbard.
The results concerning the snow budgets, the wet deposition, and the ratios of the different components in the
snow pack, in the precipitation, and in the aerosols underline the importance of wet deposition for the major sea-





salt components chloride, sodium, potassium, magnesium, and sulfate during the winter period from October
2011 to March 2012 confirming previous studies (Isaksson et al., 2003; Weinbruch et al., 2012; Geng et al.,
2010). The significant contribution of wet deposition is further supported by the estimated upper limit of the dry
aerosol deposition. Although the choice of the deposition velocity introduces considerable uncertainty, the
estimated dry deposition remains well below 10 % of the total deposition for chloride, sodium, and magnesium,
while it contributes more than 20 % to the snow budget of calcium and potassium probably due to a stronger dust
contribution. It is possible that the relatively high overall accumulation including strong precipitation events in
the last week of January contributed to the high input due to wet deposition during the winter 2011/2012.
Therefore, the contribution of dry deposition of sea salt aerosols could be larger during winter periods with
different precipitation characteristics. Nevertheless, it appears that the wet deposition measurements at Ny-
Ålesund can be used to estimate the total wintertime deposition of the major sea salt components in the areas
surrounding Ny-Ålesund.
In contrast to the major sea salt components the dry deposition of nitrate and nss-sulfate was more important
than the wet deposition. However, the dry deposition of the corresponding gas phase species like $HNO_3$ and $SO_2$
are not well quantified (e.g. Zhang et al., 2001; Osada et al., 2010) and probably contributed also to the observed
snow budgets of nitrate and sulfate. Further direct measurements of the dry deposition of all N-containing
species and nss-sulfate would be needed to better quantify the full N- and S-cycle in the Arctic.
The obtained results for the snow budgets and the deposition of BC indicate a behavior of BC resembling nitrate
and nss-sulfate. In the wintertime the deposition of BC to the snowpack on the glaciers surrounding Ny-Ålesund
appears to be equally driven by dry and wet deposition. However, it is important to note the large uncertainties in
the estimated BC deposition, for which direct measurements in the Arctic are needed. Overall, the average rBC
concentrations in the wintertime snowpack remained below 1.2 ppb and, thus, causing a marginal reduction of
the snow albedo. In contrast, post-depositional processes are likely at the origin of snow layers with rBC
concentrations increased by a factor of three compared to the average. Such layers may cause a stronger direct
and indirect impact on the snow albedo via enhanced metamorphism processes (e.g. Jacobi et al., 2015). Further
studies with detailed observations of the vertical BC distribution in the snow pack are required for a better
quantification of the climate impact of BC in snow.
Bromide is the sea salt compound showing the strongest variability in the ratio to other major components like
sodium, which is related to its high mobility in the sea ice-atmosphere-snow system caused by chemical
processes. Systematic measurements of bromide not only in the snowpack, but also in the aerosols, in the
precipitation, and in fresh snow are required to further investigate processes before the formation of the sea salt
aerosols, during their transport, or after the deposition to the snow pack.
While the annual budgets and estimated deposition for most of the studied species agree well, the results for the
monthly budgets obtained with the detailed snowpack modeling are less convincing. Further improvements
regarding the modeling of the Arctic snow pack are needed to better address physical properties (e.g. the
evolution of the snow density) and post-depositional processes acting upon the vertical distribution of impurities
in the snow pack. Although the treatment of impurities was recently implemented into the Crocus snowpack
model (Tuzet et al., 2017), the impact of processes modifying the vertical distribution of impurities in the Arctic
snowpack like blowing snow, sublimation, and percolation are still not fully considered in most models. The full



implementation of post-depositional processes into complex snow models may offer the opportunity to exploit
further snow pack and ice core observations for the reconstruction of climate and pollution.

*Code and data availability*. The snowpack scheme Crocus is integrated into the surface modeling platform
SURFEX developed by Météo-France. The SURFEX code is freely available via www.umr-cnrm.fr/surfex/
using a CECILL-C license. The snow pit (https://doi.org/10.6096/parcs.12, https://doi.org/10.6096/parcs.13) and
the meteorological data (https://doi.org/10.6096/parcs.17) are available from the PARCS data base (www4.obs-
mip.fr/parcs/database/). Precipitation data are available from the EBAS data base of the Norwegian Institute for
Air Research (ebas.nilu.no). BC data is available on request.

*Author contributios*. HWJ performed the snow sampling, the simulations, and the analysis and wrote the
manuscript. FO provided meteorological data for the simulations, advice during the analysis, and support in
writing the manuscript. SDC contributed to the snow sampling and analysis. PG performed the chemical analysis
of the snow samples. KE provided atmospheric BC data. WA provided precipitation data. PG, KE, and WA
contributed to the writing of the manuscript. MZ contributed to the snow sampling, performed the BC analysis of
the snow samples, and provided support in writing and designing the manuscript.

*Competing interests*. The authors declare that they have no conflict of interest.

*Acknowledgments*. The research project No. 1030 (CLIMSLIP-NyA) was performed at the AWIPEV Station. It
was supported by the French Polar Institute (IPEV), the Agence Nationale de la Recherche under the contract
ANR 2011 Blanc SIMI 5-6 021 04, by Campus France (No. 31597SM) and the Austrian Science Fund (FWF,
grant I 369-B17), by the Chantier Arctique Francais (CNRS/INSU) via the project PARCS, and by the Labex
OSUG@2020 (Investissements d'avenir – ANR10 LABX56). MZ gratefully acknowledges the funding by the
Deutsche Forschungsgemeinschaft (DFG, German Research Foundation) – Projektnummer 268020496 – TRR
172, within the Transregional Collaborative Research Center "ArctiC Amplification: Climate Relevant
Atmospheric and SurfaCe Processes, and Feedback Mechanisms (AC)³. The KV weather station is operated in
cooperation with the Norwegian Polar Institute (Tromsø). The measurements of major inorganic ions in
precipitation and aerosols at Ny Ålesund and Zeppelin are part of the national atmospheric monitoring financed
by the Norwegian Environment Agency.

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
