# Peer review of "Deposition of ionic species and black carbon to the Arctic snow pack: Combining snow pit observations with modeling"

_Atmospheric Chemistry and Physics, 2019_

## Referee Comment (RC1) · Anonymous Referee #1 · 10 May 2019

In this paper the authors are combining a variety of field observations data from Ny Ålesund (Svalbard) with modelling in order to improve the understanding of aerosol removal and deposition in the Arctic. For being in the Arctic Ny Ålesund has a large variety of observational data available and thus this is a suitable site for such a study. The papers describes the data, methods, model and conclusions very clearly and it is a generally well written providing important knowledge of in particular sea salts, nitrate, nss sulfate and BC deposition to the winter snowpack. However, before the paper is accepted for publication I would like to see the following aspects better explained. Data from two snow pits from two glaciers are used in this study. The glaciers are both situated in the Kongsfjorden area but are fundamentally very different. Austre Lovenbreen

is a low elevation, small valley glacier where the surrounding topography are affecting both the precipitation distribution (wet deposition) and wind and turbulence (dry deposition). The other glacier, Kongsvegen, is much larger, has a different elevation distribution and thus deposition and precipitation distribution. What was the reason for selecting these particular sites?

In addition to the fundamental differences between the two glaciers I also wonder about the spatial variability at each of these sites? How representative is one snow pit for this particular glacier?

Would the elevation difference between the two snow pits sites, Zeppelin and Ny Ålesund have any effect on the conclusions from this study?

Is it possible that the surface height changes recorded by the ultasonic ranger is affected by rain and/or windblown snow and giving misleading results regarding precipitation events?

The citation of relevant papers is fine but I do miss the mentioning of one important study; one of the landmark studies regarding BC in Ny Ålesund which also presents observational evidence of dry deposition on higher elevation glacier snow- something that is not often seen in the Arctic. Stohl, A., et al. Arctic smoke – record high air pollution levels in the European Arctic due to agricultural fires in Eastern Europe in spring 2006, Atmos. Chem. Phys., 7, 511-534, https://doi.org/10.5194/acp-7-511-2007, 2007.

Finally, I would like to congratulate the authors to an interesting and well-written study!

---

## Referee Comment (RC2) · Anonymous Referee #2 · 16 May 2019

Overall assessment:

This paper describes a detailed study of sea salt ions and black carbon (BC) deposition on two glaciers of northern Svalbard, and combines snow pit observations with aerosol measurements at Zeppelin station to estimate the relative contributions of wet vs. dry deposition to the observed budgets of aerosols in the snow pack. The paper addresses an existing knowledge gap with parallel measurements of airborne BC concentrations and accumulation in snow, which helps clarify the processes governing the air-to-snow transfer of these aerosols which are susceptible to affect the surface radiative balance, and hence polar climate. The contribution is therefore valuable, and, although not

ground-breaking or highly original, the paper is of sufficient interest to be published in ACP. Overall, the paper is well written, easy to read, neither too wordy nor too brief, and it adequately cites recent and relevant scientific literature. The figures are likewise simple, clear and easily understandable. The choice of methods is appropriate and good care appears to have been taken at most, if not all, steps of the various analyses to ensure results of quality. Sufficient information is provided to justify choices in the various data and parameters used in calculations. The main conclusions are also well supported by the data presented. I do, however, find some lacunae in some part of the method descriptions, as well as some ambiguities in a few parts of the manuscript. These need to be addressed before the paper can be accepted.

Specific questions and comments:

Section 2.1 (Snow sampling) Please explain the snow pit sampling strategy more clearly: Was sampling done at fixed depth increments, or within each stratigraphically distinct layer, or a combination of both ?

Section 2.2 (Snow analysis) How much type elapsed between samples collection, melting, and analysis for ions and for BC ? Please provide more information about the SP2 analyses. Some of this info can be placed in the Supplement. Show the Aquadag calibration curves. You report a mean aerosolization efficiency of only 56 %, much lower than the 72 % reported by Lim et al. (2014) with the same nebulizer. How come ? How variable was the efficiency between analytical runs ? Did you estimate the nebulization efficiency as a function of particle size, or just assumed it to be as described in Lim et al. (2014) ? If you did estimate the efficiency for different particle sizes, show the results. Otherwise, do you have any results from analyses performed on other samples (e.g., ice cores) with the same instruments that could be used to quantify these sources of uncertainty ? This is important since detection efficiency on the SP2 is often lowest for the largest particles that hold moss of the BC mass.

Section 2.6 (Dry and wet deposition) L172: Add the reference (Millero et al.) for the

standard seawater ionic ratios.

Section 3.2 (Impurity profiles in the snow pack) Please clarify what exactly the "adjustments" in the snow pit impurity profiles (Fig. 3) actually mean ? What was adjusted, and how ? This is not really clear.

Section 3.3 (Wintertime snow budgets and deposition of ionic compounds) L315: How was the 20 % figure estimated ? This comes across as a pure guess. Explain more clearly.

Section 3.4 (Wintertime snow budgets and deposition of BC) L355-356: Some of the differences discussed in this paragraph (with the data of Forsström et al.) are as likely or more likely to be due to the differences in methods (SP2 vs TOT) than to spatial variability.

Section 3.5 (Comparison of monthly snow budgets and deposition) L405: What model and forcing data are you referring to ? This sentence is unclear. L409-410: I disagree with this statement: I see no convincing evidence on Fig. 5 that there is a better agreement between the measured and calculated monthly snow pit budgets of nitrate and BC than for sodium. If you have some metrics to support this otherwise vaguely qualitative statement, provide them.

Section 3.6 (Variations of the chemical composition...)

L461-468: $R^2$ is not the correlation coefficient, it is the coefficient of determination. If $R = 0.60$, $R^2 = 0.36$, i.e. only 36 % of the variability is explained by the linear model. You also state that the average ratio of rBC to nssSO4 in snow is almost an order of magnitude than in the atmosphere. I presume that by the "atmosphere" you mean the aerosols at Zeppelin ? (shown on Fig. 7), if so please clarify the sentence. Also, please show the equations of the regression lines, so that the reader can actually compare the mean ratios (slopes).

Section 4 (Conclusion)

L538-539: Explain how you conclude that 1.2 ppb of rBC caused a marginal reduction of snow albedo. Provide a reference.

igures:

Fig. 2, 3, 8: I suggest adding small labels on the edge of the graphs, identifying the monthly subdivisions of the snowpack, for greater clarity.

Fig. 8: I suggest adjusting the x-axis scale on both panels to the same min-max range as on the Konsvegen panel (left-hand). The few large peaks in Br/Na on the Austre Lovenbreen panel (right-hand) obscure the variations of the baseline values, and makes the comparison between the two panels difficult. The peaks can be truncated and the peak values shown in labels.

See PDF attachment for suggested minor changes in wording in some sentences.

Please also note the supplement to this comment:
https://www.atmos-chem-phys-discuss.net/acp-2019-215/acp-2019-215-RC2-supplement.pdf

**Supplement:**

**Review of "Deposition of ionic species and black carbon to the Arctic snow pack (...)"**
**by H.W. Jacobi *et al.* (ACP manuscript acp-2019-215)**

**Suggested wording changes:**

L136: "The model is forced using meteorological data,  including air, temperature (...)"

L227-228: "Therefore,  some redistribution of the impurities probably occured in the snow pack due to melting. This is however unlikely to have led to complete  elution.  Hence, the overall impurity budgets of both snowpits  are assumed to have been mostly unaffected by melt.

L357: The monthly averages (...)

L363:  Noone and Clarke (1988) proposed...

L422-423: Most of the chloride-to-sodium ratios in the aerosols are close to or less than the standard sea water ratio (...)

L444: ....as well as in the precipitation, which  results in calcium-to-sodium ratios (...)

L451-453: On average, the highest  ratios are found in  the precipitation and the lowest in the aerosols (...)

L460-461: (...) atmospheric BC shows a  positive relationship to nss-sulfate (...)
        *[note: the linearity is weak]*

L519: (...) is further supported by the estimated  maximum  dry  deposition.

---

## Author Comment (AC1) · 1 Jul 2019

*Our responses to the reviewers' comments are written in italic.*

**Referee # 1**

Data from two snow pits from two glaciers are used in this study. The glaciers are both situated in the Kongsfjorden area but are fundamentally very different. Austre Lovenbreen is a low elevation, small valley glacier where the surrounding topography are affecting both the precipitation distribution (wet deposition) and wind and turbulence (dry deposition). The other glacier, Kongsvegen, is much larger, has a different elevation distribution and thus deposition and precipitation distribution. What was the reason for selecting these particular sites?

*Despite technological progress and better access to polar regions, research in these areas is still limited by logistic constraints and meteorological conditions. During the design of the project, we planned snow pits in the vicinity of the village of Ny-Alesund to be as close as possible to the supporting measurements in the village and at Zeppelin Station. However, exceptional meteorological conditions during the winter 2011/2012 with large rain events in January caused a melting of the entire snowpack at low altitudes, which turned into a thick layer of ice in the following days. This layer was not exploitable concerning chemical budgets. Since the snowpack remained intact at higher altitudes on the surrounding glaciers, we were forced to shift our sampling to such sites. The most obvious choice for the first snow pit was the Kongsvegen glacier equipped with automatic weather stations providing the required meteorological data for the forcing of the snowpack model. A second snow pit on the same glacier at a different weather station appeared less interesting to us than a snow pit on a second glacier.*

*For this second snow pit the Austre Lovenbren was in our opinion the best choice since it is one of the best studied glaciers close to Ny-Alesund with field observations reaching back to the 1960ies (Merlin, C., et al., Change in Geometry of a High Arctic glacier from 1948 to 2013 (Austre Lovénbreen, Svalbard), Geografiska Annaler A Phys.Geogr. 99 (2), 115-138, dx.doi.org/10.1080/ 04353676.2017.1285203, 2017). In addition, while the reviewer's observations are valid for the slopes, the central part of the glacier, where the snow pit was located, is in a more protected area. Field observations indicate a rather homogeneous distribution of the snow cover in the central part only slightly influenced by orography or wind (Bernard, E., et al., Monitoring seasonal snow dynamics using ground based high resolution photography (Austre Lovénbreen, Svalbard, 79°N), ISPRS J.Photogramm.Rem.Sens. 75, 92-100, doi.org/10.1016/ j.isprsjprs.2012.11.001, 2013; Bernard, E. et al., Investigating snowpack volumes and icing dynamics in the moraine of an Arctic catchment using UAV/photogrammetry, Photogramm.Rec. 32 (160), 497-512, doi.org/10.1111/phor.12217, 2017). Therefore, we opted for a second snow pit on this glacier to investigate the variability of the snow budgets not only concerning altitude, but also concerning different glaciers. We further hope that our results potentially provide valuable input for the studies on the Austre Lovenbren.*

*Nevertheless, we agree with the reviewer that the description of the glacier properties are possibly not detailed enough and we propose to add the following information about the glaciers in chapter 2.1:*

*"The Kongsvegen glacier is one of the largest glaciers in the vicinity of Ny-Alesund with an area of ~100 km² and a length of 26 km. It reaches from sea level to a maximum altitude of 750 m with slopes ranging from 0.5 to 2.5° (Karner et al., 2013). Austre Lovenbreen is a low elevation, small valley*

*glacier with an area of ~10 km² with a surrounding topography affecting meteorological conditions like wind and precipitation. Nevertheless, the snow pit was located in the central part of the glacier with rather homogeneous snow properties (Bernard et al., 2017)."*

In addition to the fundamental differences between the two glaciers I also wonder about the spatial variability at each of these sites? How representative is one snow pit for this particular glacier?

*Spatial variability of chemical concentration in the snow has been addressed in several previous studies. The variability can play a role at different scales from cm to km. Actually, using two snow pits on two different glaciers addresses the variability at a scale of several 10s of km. In this case the results show rather homogeneous snow budgets despite differences in altitude (Fig. 4). Concerning the small scale variability previous studies showed variations for example of BC in snow on the order of 20 % at a scale of several meters, which may increase significantly due to blowing snow (Svensson et al., 2013 and references therein). We estimate here an overall uncertainty of 21 % due to error propagation from the combination of the spatial variability of 20 % (Svensson et al., 2013), the analytical error of 5 %, and the error of the density measurements of 6 % (Proksch et al., 2016). As a result all calculated (total and monthly) budgets differing by less than 21 % are not considered to be significantly different.*

*We propose to rectify in the revised manuscript the value of 20 % to determine significantly different budgets to 21 % according to the above described calculations. Moreover the brief description for the justification of our previous estimate in ch. 3.3, lines 314 to 317 will be replaced by the description of the above mentioned calculated uncertainty, which will be added to chapter 2.4.*

Would the elevation difference between the two snow pits sites, Zeppelin and Ny Ålesund have any effect on the conclusions from this study?

*The elevation of the sampling sites vary between ~5 m a.s.l. (precipitation, Ny-Alesund), ~475 m a.s.l. (aerosols, Zeppelin), and ~340 m a.s.l. and ~670 m a.s.l. (snow pits). Apparently, it is possible that wet and dry deposition are not directly comparable at these sites.*

*For example, the wet deposition determined at sea level may be impacted by below-cloud scavenging in the altitude range between the elevation of the snow pits and sea level. If this was the case, the calculated wet deposition at Ny-Alesund may be higher than the snow budgets. However, the results of most compounds give no indication of a detectable role of below-cloud scavenging in the elevation range between the snow pits and sea level. As discussed in the manuscript (l. 443ff) this is potentially not the case for calcium, for which below-cloud scavenging of soil-generated dust may occur close to sea level, but not over the glaciers. The higher wet deposition of bromide compared to the snow budget is most probably related to a positive bias of wet deposition, which was estimated using standard sea water ratios of bromide-to-sodium due to the lack of direct bromide measurements in the precipitation.*

*The dry deposition estimated from the Zeppelin measurements may be impacted in case of strong vertical aerosol concentration gradients. This adds to the uncertainty of the estimated dry deposition,*

*which also suffers from the simplified approach according to equation (1). Nevertheless, due to the relatively small contribution of dry deposition for most of the examined compounds, the uncertainties in dry deposition do not affect the conclusions for these compounds. The impact of vertical aerosol concentration gradients is obviously larger for nss-sulfate, nitrate, and BC with larger estimated contributions of dry deposition. Unfortunately, information on such gradients is missing for nitrate and nss-sulfate. Only in the case of BC some studies have been performed as described in the manuscript. These studies deliver mixed results from homogeneous atmospheric distribution to strong differences mainly due to the layering of different air masses (e.g. Spolaor et al.; 2017). While these different gradients may have had compensating effects for the overall budgets during the period covered by the snow pits, they may as well have contributed to the observed differences in the BC profiles in the two snow pits (Fig. 3c and d). We propose to add to ch. 3.2, first paragraph the following sentence: "These differences may be caused by site-dependent deposition fluxes. For example, strong vertical gradients in aerosols concentrations between the altitudes of the snow pits as sometimes observed for BC (Spolaor et al., 2017) have a direct impact on the local deposition fluxes."*

Is it possible that the surface height changes recorded by the ultasonic ranger is affected by rain and/or windblown snow and giving misleading results regarding precipitation events?

*The used sensor (SR50a, Campbell Sci.) has a nominal accuracy of ±1 cm and a scan takes about 1 s including an independent temperature measurement used for the corrections of variations in sound propagation. Indeed, the recorded signals can be disturbed by particles moving through the sensor's field of view (30°). This is less a problem with rain and snow fall occurring at lower wind speeds. If, however, precipitation is accompanied by high wind speeds, the horizontally moving drift particles can induce noisy signals. Note, that the sensor principally responds to the closest object in its field of view, but signals are rejected if targets move with a velocity beyond 4 cm s$^{-1}$, which can result in missing records during snow drift. Signals passing this check can still be quite noisy and, therefore, require some post processing to filter for significant values. In the current analysis we first applied the temperature correction and then only considered data showing an increase of surface height larger than 1 cm h$^{-1}$ at temperatures below +1 °C and relative humidity larger than 95 %. These data were converted to water equivalent based on a wind and temperature dependent parameterization of fresh snow density implemented in the CROCUS snow model (Vionnet et al., 2012). A factor of 1.1 was applied to these values to finally ensure consistency with total snow height and density observed in investigated snow pits. Overall, this procedure yields data of snow accumulation and, thus, represents changes in water equivalent in response to precipitation and subsequent snow redistribution. The data are prone to uncertainties during snowfall events themselves, but the timing and the net changes at the end of major precipitation events as well as their total at the end of the winter season are considered reliable.*

*We propose to modify ch. 2.3 adding the following text:*

*"Temperature corrected raw data were filtered for changes of more than 1 cm hr-1 to filter significant events and exclude potentially inaccurate data, which also may be due to potential disturbances by drifting snow. The significance of these events was crosschecked by temperature and humidity criteria. The resulting data were converted to water equivalent using a wind and temperature*

*dependent parameterization of fresh snow density as used in the CROCUS snow model (Vionnet et al., 2012). A scaling factor of 1.1 was applied to achieve consistency with total snow height and average density observed in investigated snow pits (Sauter and Obleitner 2015). Overall, this procedure yields data of snow accumulation and, thus, represents changes in water equivalent in response to snow precipitation and subsequent redistribution. More details about this procedure as well as about quality control of further recorded meteorological parameters are given in Karner et al. (2013) and Sauter and Obleitner (2015)."*

The citation of relevant papers is fine but I do miss the mentioning of one important study; one of the landmark studies regarding BC in Ny Ålesund which also presents observational evidence of dry deposition on higher elevation glacier snow- something that is not often seen in the Arctic. Stohl, A., et al. Arctic smoke – record high air pollution levels in the European Arctic due to agricultural fires in Eastern Europe in spring 2006, Atmos. Chem. Phys., 7, 511-534, https://doi.org/10.5194/acp-7-511-2007, 2007.

*The paper by Stohl et al. is certainly an important paper describing an exceptional pollution event at Ny-Alesund in early May 2006 including a wealth of atmospheric observations. However, for the atmosphere-snow interaction including deposition only circumstantial and non-quantitative evidence is presented including a photograph of the surface with discolored snow and the analysis of a small number of surface snow samples taken on the Holtedahlfonna glacier. While the analysis clearly indicates elevated concentrations of ionic compounds in the discolored snow (by the way, the snow samples were not analyzed for BC), Stohl et al. state that "dry deposition is likely a too slow process to cause the discoloration of the snow within hours, …". Nevertheless, since they also mention blowing snow as potential mechanism for the enhancement of the deposition, we propose to modify the discussion of the deposition in the introduction in the revised manuscript according to:*

*"Moreover, Stohl et al. (2007) evoked a potentially large enhancement of the deposition of impurities due to blowing snow during a period with elevated aerosol concentrations."*

**Referee # 2**

Section 2.1 (Snow sampling) Please explain the snow pit sampling strategy more clearly: Was sampling done at fixed depth increments, or within each stratigraphically distinct layer, or a combination of both?

*The strategy consisted in sampling all visible snow layers at least once if possible. This was realized in both snow pits. However, in some cases it was impossible to sample a distinct layer due to its limited thickness and/or its elevated hardness. We propose to add to ch. 2.1:*

*"If allowed by the thickness and the hardness, each identified snow layer was sampled separately and at least once."*

Section 2.2 (Snow analysis) How much type elapsed between samples collection, melting, and analysis for ions and for BC?

*We propose to add the following information to ch. 2.2: "All samples were melted in a thermostated water bath and were analyzed within 30 min after melting."*

Please provide more information about the SP2 analyses. Some of this info can be placed in the Supplement. Show the Aquadag calibration curves. You report a mean aerosolization efficiency of only 56 %, much lower than the 72 % reported by Lim et al. (2014) with the same nebulizer. How come? How variable was the efficiency between analytical runs? Did you estimate the nebulization efficiency as a function of particle size, or just assumed it to be as described in Lim et al. (2014)? If you did estimate the efficiency for different particle sizes, show the results. Otherwise, do you have any results from analyses performed on other samples (e.g., ice cores) with the same instruments that could be used to quantify these sources of uncertainty? This is important since detection efficiency on the SP2 is often lowest for the largest particles that hold moss of the BC mass.

*The aerosolization efficiency was estimated from the mass only and controlled on a daily basis. No size dependent correction was applied. On the first two days of measurements an error in the dilution of the mother Aquadag suspension resulted in wrong efficiency factors, which were higher than 100 %. For such reasons, the 56 % aerosolization efficiency was calculated with the remaining four daily calibrations. Considering the low day-to-day variability (53-60%), the averaged aerosolization efficiency was applied to all samples.*

*The discrepancy of the aerosolization efficiency reported in the present work and in the work of Lim et al. (2014) is somewhat surprising. The analyses of the different samples were performed within only several months, but with modifications to the sampling line. However, we cannot provide a clear explanation for these different efficiencies. Nevertheless, this change shows the importance of daily calibrations and test of the nebulizer prior to the analysis of real samples.*

*While we agree that such additional information on the SP2 measurements can be beneficial, we believe that they are of interest only to a small fraction of the readers of the paper. Therefore, we propose to add the following paragraph to the Supplementary Material:*

*"Additional information for the SP2 measurements*

*The aerosolization efficiency of the nebulizer was estimated only from the mass and without a size-dependent correction. The efficiency was determined on four out of six days of measurements indicating a day-to-day variability between 53 and 60 % (Fig. S1, left). Considering the low day-to-day variability, the calculated average aerosolization efficiency of 56 % (Fig. S1, right) was applied to all samples.*

[Figure]

*Fig. S1: (Left) Measurements of the aerosolization efficiency of the nebulizer using Aquadag suspensions on four days (calib #1 to #4). (Right) Statistics for the calculated aerosolization efficiency for each day and for all measurements."*

Section 2.6 (Dry and wet deposition) L172: Add the reference (Millero et al.) for the standard seawater ionic ratios.

*We will move upward the reference to Millero et al., which appears in the same paragraph in the presentation for the wet deposition of bromide, to the explanation for the dry deposition of bromide in ch. 2.6.*

Section 3.2 (Impurity profiles in the snow pack) Please clarify what exactly the "adjustments" in the snow pit impurity profiles (Fig. 3) actually mean? What was adjusted, and how ? This is not really clear.

*We combined stratigraphic and chemical information to derive full chemical profiles from the measured concentrations, which each cover only a vertical range of 3 cm. We adjusted the concentrations by assuming homogenous concentrations in an identified layer and extending the observed concentrations to the top and bottom of the concerned sow layer.*

*We propose to add at the beginning of ch. 3.2 the following clarification: "Profiles were established by assuming homogeneous concentrations for the identified snow layers and by adjusting the closest observed concentrations to the vertical extent of the observed or simulated layers."*

Section 3.3 (Wintertime snow budgets and deposition of ionic compounds) L315: How was the 20 % figure estimated? This comes across as a pure guess. Explain more clearly.

*See reply to remark # 2 of referee # 1.*

Section 3.4 (Wintertime snow budgets and deposition of BC) L355-356: Some of the differences discussed in this paragraph (with the data of Forsström et al.) are as likely or more likely to be due to the differences in methods (SP2 vs TOT) than to spatial variability.

*We stated already in ch. 3.2: "It should be noted here and during all further discussion, that the comparison of the different quantities EC, eBC, and rBC introduces additional uncertainties." We propose to recall this information in ch. 3.4 by modifying the last sentence of the first paragraph to: "These differences, which are partly related to the different detection methods for BC, are probably also related to the inter-annual variability of BC as reflected in the atmospheric observations (Eleftheriadis et al., 2009). "*

Section 3.5 (Comparison of monthly snow budgets and deposition) L405: What model and forcing data are you referring to? This sentence is unclear.

*This sentence concerns the snow modeling and the corresponding forcing data. We propose to modify the sentence to " ...caused by multiple reasons related to uncertainties in the snow model results, the corresponding forcing data as well as in the spatial variability of the observations..."*

L409-410: I disagree with this statement: I see no convincing evidence on Fig. 5 that there is a better agreement between the measured and calculated monthly snow pit budgets of nitrate and BC than for sodium. If you have some metrics to support this otherwise vaguely qualitative statement, provide them.

*We suggest removing the concerned sentence.*

Section 3.6 (Variations of the chemical composition...) L461-468: $R^2$ is not the correlation coefficient, it is the coefficient of determination. If R = 0.60, $R^2$ = 0.36, i.e. only 36 % of the variability is explained by the linear model. You also state that the average ratio of rBC to nssSO4 in snow is almost an order of magnitude than in the atmosphere. I presume that by the "atmosphere" you mean the aerosols at Zeppelin? (shown on Fig. 7), if so please clarify the sentence. Also, please show the equations of the regression lines, so that the reader can actually compare the mean ratios (slopes).

*The value of 0.6 corresponds to the coefficient of determination as pointed out by the reviewer. The text as well as Fig. 7 refers to aerosols. This will be rectified in the revised manuscript. The values for the slopes and the coefficients of determination will be added to Fig. 7.*

Section 4 (Conclusion) L538-539: Explain how you conclude that 1.2 ppb of rBC caused a marginal reduction of snow albedo. Provide a reference.

*In a previous modeling studies we showed that the broad-band snow albedo reduction due to 100 ppb of BC corresponds to ~0.03 (Jacobi et al., 2015), which matches a reduction of ~4 % of a snow albedo*

*of 0.8. Wiscombe and Warren found that 100 ppb of BC reduces the albedo at a wavelength of 0.4 µm by 1 %. The extrapolation of this values leads to an albedo reduction smaller than 0.05 % in the presence of 1.2 ppb of BC in the snow. We propose to add the corresponding references to the manuscript.*

Figures: Fig. 2, 3, 8: I suggest adding small labels on the edge of the graphs, identifying the monthly subdivisions of the snowpack, for greater clarity.

Fig. 8: I suggest adjusting the x-axis scale on both panels to the same min-max range as on the Konsvegen panel (left-hand). The few large peaks in Br/Na on the Austre Lovenbreen panel (right-hand) obscure the variations of the baseline values, and makes the comparison between the two panels difficult. The peaks can be truncated and the peak values shown in labels.

*The proposed modifications of the figures will be done.*

See PDF attachment for suggested minor changes in wording in some sentences.

*We thank the reviewer for the careful reading of the manuscript and the suggested changes, which were mostly taken into account.*